# Contribution of Essential Oils to the Fight against Microbial Biofilms—A Review

**Diana Camelia Nuță** [1,*], **Carmen Limban** [1], **Cornel Chiriță** [2], **Mariana Carmen Chifiriuc** [3], **Teodora Costea** [4], **Petre Ioniță** [5], **Ioana Nicolau** [5] and **Irina Zarafu** [5]

1   Department of Pharmaceutical Chemistry, Faculty of Pharmacy, "Carol Davila" University of Medicine and Pharmacy, TraianVuia no.6, 020956 Bucharest, Romania; carmen.limban@umfcd.ro
2   Department of Pharmacology and Clinical Pharmacy, Faculty of Pharmacy, "Carol Davila" University of Medicine and Pharmacy, TraianVuia no.6, 020956 Bucharest, Romania; cornel.chirita@umfcd.ro
3   Department of Microbiology, Faculty of Biology, Universtity of Bucharest, AleeaPortocalelor no.1-3, 060101 Bucharest, Romania; carmen_balotescu@yahoo.com
4   Department of Pharmacognosy, Phytochemistry, Phytotherapy, Faculty of Pharmacy, "Carol Davila" University of Medicine and Pharmacy, TraianVuia no.6, 020956 Bucharest, Romania; teodora.costea@umfcd.ro
5   Department of Organic Chemistry, Biochemistry and Catalysis, Faculty of Chemistry, University of Bucharest, Regina Elisabeta no.4-12, 030018 Bucharest, Romania; petre.ionita@chimie.unibuc.ro (P.I.); ioana.nicolau@chimie.unibuc.ro (I.N.); irina.zarafu@chimie.unibuc.ro (I.Z.)
*   Correspondence: diana.nuta@umfcd.ro

**Abstract:** The increasing clinical use of artificial medical devices raises the issue of microbial contamination, which is a risk factor for the occurrence of biofilm-associated infections. A huge amount of scientific data highlights the promising potential of essential oils (EOs) to be used for the development of novel antibiofilm strategies. We aimed to review the relevant literature indexed in PubMed and Embase and to identify the recent directions in the field of EOs, as a new modality to eradicate microbial biofilms. We paid special attention to studies that explain the mechanisms of the microbicidal and antibiofilm activity of EOs, as well as their synergism with other antimicrobials. The EOs are difficult to test for their antimicrobial activity due to lipophilicity and volatility, so we have presented recent methods that facilitate these tests. There are presented the applications of EOs in chronic wounds and biofilm-mediated infection treatment, in the food industry and as air disinfectants. This analysis concludes that EOs are a source of antimicrobial agents that should not be neglected and that will probably provide new anti-infective therapeutic agents.

**Keywords:** essential oils; bacterial biofilm; antimicrobial; medical devices

## 1. Introduction

In the past decade, essential oils (EOs) use for the prophylaxis and therapy of biofilm-associated infections (BAIs) have become very popular. The universally accepted definition of a biofilm refers to a sessile multicellular community of microbial cells with a modified transcriptome and phenotype (exhibiting increased resistance to both therapeutic doses of current antimicrobials and immune effectors) that adhere to a surface and, among them, being protected by an auto-secreted extracellular polymeric matrix [1,2]. It is considered that BAIs represent up to 85% of the total microbial infections, occurring after microbial colonization of either viable tissues or medical devices and having serious consequences [3], because they are persistent and hard or impossible to treat, even in immunocompetent individuals.

On the other hand, the food manufacturing industry is facing the formation of microbial biofilms that can affect industrial processes, and researchers are in constant search of new ways to eradicate this phenomenon. The protective mechanisms of microbial cells within biofilms are multifactorial and differ from those that occurred in planktonic cells and include matrix impermeability, modified transcription rate, selection of persister cells,

accumulation of antibiotic inactivating enzymes, increased horizontal transfer rate of resistance genes, etc. [4]. Therefore, biofilm cells can become up to100–1000 times more resistant to antimicrobial substances than planktonic cells [5]. This high phenotypic resistance, also called tolerance, interferes not only with the BAI treatment but also with the efficacy of surface disinfection processes [6,7]. The discovery of natural products with antimicrobial activity represents a direction of current research trying to limit microbial diseases.

Therefore, the purpose of this paper was to review the recent literature on EOs antibiofilm activities.

## 2. Methods

For this purpose, the PubMed (National Library of Medicine, Washington, DC) and Embase (Elsevier) databases were searched for all relevant articles written in English, using the following keywords: "essential oils", and "biofilm", and then "dentistry", "chronic wound infections", "medical devices", "food industry". We also reviewed additional relevant articles identified from the referenced citations. We limited our investigation to English-language journals.

EOs (also called volatile or ethereal oils) are natural aromatic oily liquids with complex compositions obtained from different plant organs by various methods, including expression, fermentation, enfleurage and extraction. The most used technique is steam distillation [8]. From the 20–60 low molecular weight components, which can be found in the composition of EOs in different amounts, at least one, such as terpenes and terpenoids or other aromatic compounds, exhibit antimicrobial activity [9] (Figure 1).

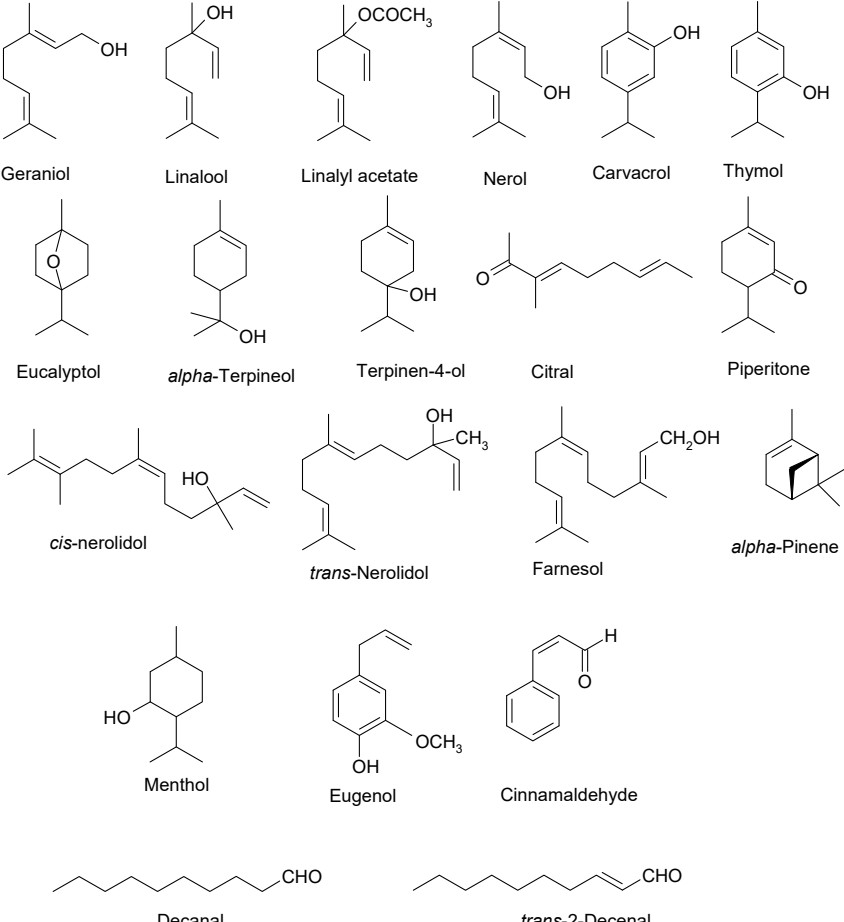

**Figure 1.** The main components of essential oils (EOs).

EOs of vegetal origin have been used for millennia in ethnomedicine as a natural antimicrobial and antiviral agents. Their antimicrobial activity is due to the alteration of microbial cell envelope leading to cellular lysis with cell contents leakage and proton motive force inhibition. Their benefits result from the fact that microbial resistance is less probably to be installed, as compared to chemical substances, the facile preparation, high biocompatibility and biodegradability [6]. Besides their antimicrobial activity, EOs also exhibit anti-inflammatory effects. In addition, their antibiofilm activity came to researchers' attention in the last ten years [4].

## 3. Discussion

### 3.1. EOs Mechanism of Action as Antibiofilm Agents

The complex composition of EOs suggests that multiple mechanisms, probably acting synergically, are involved in their biological effects [3]. From the studied articles, we identified several types of mechanisms of action that we describe below.

In the review of Saviuc et al., it is shown that EOs are potent antibiofilm agents, acting by inhibition of the intercellular communication systems and by inducing changes in the substrate (referring to changing of redox potential, resistivity or pH) [10]. EOs could also kill the biofilm embedded cells by the alteration of the cytoplasmic membrane due to their hydrophobic constituents [11]. The research performed by Selim and Burt groups revealed that the absence of the outer membrane in Gram-positive bacteria favors the direct interaction of the EOs with the cellular membrane, either affecting its permeability and causing the leakage of intracellular content or inactivating the bacterial enzymes [8,12].

As presented by Melo et al. in terms of action on the biofilm produced by *S. aureus*, the action of EOs depends on the hydrophobicity, reactivity and diffusion rate of EOs in the matrix, as well as to the composition and structure of the biofilm. The EOs can stop the formation of biofilm by blocking the quorum-sense system, inhibiting the transcription of flagellar genes or by interfering with bacterial motility [13].

Tang et al. analyzed the EOs from *Amomum villosum* Lour against methicillin-resistant *S. aureus* (MRSA) and established that the mechanism of action consists in reducing the bacterial adherence to inert surfaces. They realized the first proteomic study of the mechanism of action of *A. villosum* Lour EO against MRSA and showed that the inhibitory effect is dose—and temperature-dependent [14].

EOs could also increase the oxidative stress in microbial cells, causing damages of intracellular macromolecules, leading to cellular apoptosis., e.g., Das et al. found that Chamomile EO induced the accumulation of reactive oxygen species (superoxide and peroxide) that could be responsible forthe antimicrobial activity of this EO [15].

In their article from 2015, Fde et al. showed that geraniol, the main constituent of some EOs, such as *Cymbopogon martini* EO, inhibits the ergosterol synthesis, a major constituent of the fungal plasma membrane [16]. *Cymbopogon citratus* EO has an effect of inhibiting glucosyltransferase activity in addition to the mechanism of degrading membrane proteins and cell permeability. This enzyme is involved in glucans synthesis, which is important for the stabilization of *E. coli* O157: H7 biofilms. The study carried out by Ortega-Ramirez in 2020 presented a type of enzymatic mechanism that essential oils can have, which can overcome the resistance to disinfection processes of microorganisms that affect the food industry, such as *E. coli* O157: H7, which can form a very resistant biofilm [17].

### 3.2. Disadvantages EOs Administration

However, the EOs disadvantages should not be neglected, as Glinel et al. have shown. The chemical composition of EOs varies, depending on the ripeness, the harvesting season and the geographical origin. The EOs have low stability when exposed to temperature or UV radiation. Some EOs are toxic after internal administration or could cause hypersensitivity reactions, dizziness, headache, nausea, or lightheadedness, mostly after skin or mucosal administration [4].

### 3.3. EOs Formulation

Particular importance presents the research on obtaining different formulations and appropriate to the use of EOs, following at the same time an accurate determination of their mode of action.

In their article from 2019, Das et al. highlighted the disadvantages of EOs, due to their increased lipophilicity and volatility, which makes it difficult the evaluate the antimicrobial effect. That's why they proposed Pickering nanoemulsion of *Chamomile* EO as a new and promising formulation, which proved to be more effective and assuring the EO intracellular delivery when tested on a wide range of bacterial and yeast strains [15].

The Pickering oil-in-water (AEP) nanoemulsion of *Artemisia annua* EO, stabilized using silica nanoparticles, was very efficient against mature *Candida* biofilms, as compared to the emulsions stabilized with Tween-80 and with the *A. annua* EO ethanolic solution, the possible mechanism of action being through the generation of the oxidative stress and superior penetration of lipid membranes, as demonstrated using the unilamellar liposomes model [18].For the same purpose, ethanol, methanol and dimethylsulfoxide (DMSO) solvents and Tween-80 or 20 surfactant-based formulations have been used to stabilize the hydrophobic EOs, reduce the volatility and improve the penetration of cellular envelope [19–22]. However, the need to obtain new formulations that increase the solubility of EO or the emulsification capacity in an aqueous medium for a more advanced release of the components still exists [22].

In the attempt to characterize the antimicrobial activity as precisely as possible, the aim was to incorporate the essential oils in modified cyclodextrins or silica nanoparticles [23–25].

Recently, inert and biocompatible solid particles have been used as stabilizing agents instead of surfactants to stabilize EO emulsions. This approach involves the adsorption of solid particles at the oil–water interface reducing the interfacial tension and thus, increasing the stability [19,26].

### 3.4. EOs as Antibiofilm Agents

At present, there are only a few compounds with demonstrated activity agents have on fungal biofilms. For this reason, new anti-biofilm molecules are needed, and some essential oils have proven effective in combating antibacterial and antifungal biofilms. The chemical composition of commonly used EOs and the biofilm-producing microbial species used in antibiofilm tests are presented in Table 1. Table 2 shows the most tested EOs in terms of the antibiofilm effect on medical devices or different surfaces.

As it is presented in Table 2, much attention has been paid lately to *Melaleuca alternifolia* (tea tree oil) (TTO EO) [27], which exhibits antibacterial and antifungal activity, preventing the formation of biofilms on different surfaces having a high-risk of contamination.

A study from 2011 performed by Budzyñska et al. presents the activity of TTO EO, *Lavandula angustifolia* (lavender essential oil) (LEO), *Melissa officinalis* (*Melissa* essential oilor lemon balm) (MEO) and linalool, linalyl acetate, α-terpineol, terpinen-4-ol on biofilms formed by *S. aureus* and *E. coli* reference strains. MEO, α-terpineol and terpinen-4-ol, showed a higher antibiofilm effect than LEO and its major components, i.e., linalool and linalyl acetate. The tests demonstrated that *E. coli* biofilm was more susceptible than the *S. aureus* biofilms to the action of EOs, especially to TTO, that destroyed it after 1 h exposure to a 0.78% concentration, contrary to the opinion stating that Gram-negative microorganisms are more resistant to EOs. In comparison with LEO and TTO, the MEO effect is more dependent on the action time. The in vivo tests on biomedical surfaces of urinary catheters and tracheal tubes showed that TTO and terpinen-4-ol used at 2 × MIC (minimal inhibitory concentration) caused visible biofilm eradication, while increased concentrations were required to eradicate the microbial biofilm on surgical mesh [5].

Karpanen and coworkers studied the antimicrobial effect of chlorhexidine digluconate, either alone or combined with TTO eucalyptus oil and thymol on planktonic and adherent *S. epidermidis*. Thymol exhibited a higher activity on biofilm cells. This was the first

study in that EOs demonstrated the best potential of inhibiting the biofilm produced by *S. epidermidis* TK1 and RP62A when they were combined with chlorhexidine digluconate. This synergistic activity is particularly valuable for skin antisepsis and for removing *S. epidermidis* from hard disinfection surfaces [28].

The study was deepened by Kwiecinski et al., who also studied the antimicrobial action of TTO; this research demonstrated the TTO effectiveness against *S. aureus* clinical strains in different growth phases, including stationary phase and biofilms. The minimum biofilm eradication concentration was usually 2x CMI, lower than 1% *v/v*. The inhibition of biofilm took place in 15 min at a TTO concentration >1% *v/v* [29].

The TTO was also tested on clinical and reference *C. albicans* strains in biofilm growth state on both biotic (human epithelial cells) and abiotic (polystyrene) surfaces, but inhibition at 0.008% concentration occurred only for one *C. albicans* reference strain [30].

In another experiment, the *Boswellia papyrifera* resin EO (BEO) proved antibiofilm activity against *S. epidermidis* and *S. aureus* preformed biofilms of 24 h, at concentrations close to the MICs. Sub-MIC concentrations of BEO exhibited a good inhibitory effect against *C. albicans* growth, adhesion, biofilm development, mature biofilm eradication (at 44 µg/mL concentration) and germ tube formation [31].

The sub-MIC concentrations of EO from *T. vulgaris* reduced the *S. aureus* biofilm development rate, inducing some cellular adaptation to this EO. This suggests that EOs treatments should envisage their rotation and combination with other biocides to prevent the emergence of resistant isolates [32].

A very recently published research on clove and thyme oils efficiency against *Fusarium* spp. development on soft contact lenses indicated that at concentrations lower than 50 µL/L that can be used without affecting the device material, the fungal cell adherence and formation were inhibited [33].

The *Mentha piperita* EO inhibits *C. albicans* and *C. dubliniensis* biofilm formation at a concentration of a maximum of 2 µL/mL in a dose-dependent manner. The effect is due to the increased concentration in menthol of this EO, which can be incorporated into the fungal cell membrane; the phenolic monoterpene, bearing a hydroxyl group on the phenolic ring, also exhibits antimicrobial effect due to the cytoplasmic membrane disruption. This mechanism of action results in the antifungal efficacy of EO of *Mentha piperita* on those strains resistant to azoles [34].

Table 1 indicates that much research has been devoted to identifying the EOs that can be used in biofilm eradication produced by *S. aureus* strains. In the study of Lee et al. (2014), from 83 EOs, nine of them (bay (*Pimenta racemosa*), cade (*Juniperus oxycedrus*), cedarwood (*Calocedrus decurrens*), frankincense (*Boswellia carterii*), lovage root (*Levisticum officinale*), oregano (*Origanum vulgare*), sandalwood (*Santalum album*), thyme red (*Thymus vulgaris*), and Vetiver Haiti (*Cymbopogon martini*) oils) inhibited the *S. aureus* biofilm at 0.01% (*v/v*) concentration. Three of them (black pepper (*Piper nigrum*), cananga (*Cananga odorata*), and myrrh (*Commiphora myrrha*) oils) at a sub-MIC exhibited a strong antibiofilm activity. One of the active compounds was cis-nerolidol (0.01% (*v/v*)), which proved to be more efficient than trans-nerolidol contained by the three EOs, inhibiting by more than 80% versus 45% the *S. aureus* biofilm growth. The black pepper EOs inhibited the expression of nuc1 and nuc2 (nucleases) and sarA (staphylococcal accessory regulator A), showing promise, together with cis-nerolidol for fighting MRSA and vancomycin–methicillin-resistant *S. aureus* infections [35].

The antimicrobial effect of carvacrol is often described in the literatureand the EOs rich in this compound, such as *Satureja hortensis* one (even in sub-MIC concentrations), proved to inhibit *Candida*, *Staphylococcus* and periodontal bacteria biofilms [36].

Gomes et al. demonstrated the efficiency of farnesol on *S. epidermidis* (one of the main nosocomial agents of indwelling medical devices BAIs) biofilm, producing significant destruction of biofilm structure and a significative reduction of biofilm thickness [37].

Gursoy et al. studied the antibiofilm activity of *Satureja hortensis* essential oil, tested on *Candida* and *Staphylococcus* biofilms, at and 0.03% and 0.06% concentrations. The growth

inhibitory effect against periodontal bacteria and the anti-biofilm effect in subinhibitory concentration was registered [36].

The EOs from two plants from the *Apiaceae* family, *Ferula asafetida* and *Dorema aucheri*, were also tested for their antibiofilm activity against *P. aeruginosa* using 25 µg/mL concentration. *Ferula* EO decreased pigmentogenesis, protease and biofilm development, while *Dorema* EO affected only pyoverdine and elastase production [38].

The EOs of *Cinnamomum burmannii* and *Massoia aromatic* are another source of antibiofilm agents, proving to inhibit both *P. aeruginosa* and *S. aureus* biofilms. The effectiveness of these EOs can be due to their main components, highlighted by GC–MS analysis, which are cinnamic aldehyde and massoia lactone, respectively [39].

Kavanaugh and Ribbeck demonstrated that the EOs of cassia, Peru balsam, and red thyme oils are very efficient against MRSA biofilms. In addition, the three EOs at MIC values have also inhibited *P. aeruginosa* biofilm cells; for cassia EO (0.2%), the effect is more intensive than that of colistin (3 µg mL$^{-1}$) [7].

The *Eucalyptus smithii* and *Juniperus communis* EOs inhibited both initial phases as well as the maturation of biofilms formed by *S. aureus* and *P. aeruginosa* respiratory isolates and reference strains [40].

A study of antibiofilm effect from 2014 used EOs from *Lamiaceae* and *Apiaceae* families (*Ammi visnaga*, *Ammoides verticillata*, *Artemisia arborescens*, *Dittrichia graveolens*, *Lavandula dentate*, *Lavandula multifida*, *Mentha piperita*, *Origanum vulgare*, *Rosmarinus eriocalyx*, *Thymbra capitata*), rich in oxygenated monoterpenes (mostly alcohols, such as thymol, carvacrol, linalool).The EOs from *T. capitata* and *O. glandulosum* (0.75–1.5%) inhibited *E. faecalis* biofilms, similar to those extracted from *A. verticillata* and *L. multifida* (1.50–3.00%). The study also confirmed that the administration of EOs is more efficient than the administration of the main component itself [41]. The *Baccharis psiadioides* (*Asteraceae*) EO, known for their antipyretic and anti-inflammatory properties, as well as a snake bites antidote, has been proved to exhibit antimicrobial and antibiofilm action on 13 *E. faecalis* resistant strains [42].

Many studies have focused on the research of eugenol, found as a major compound in clove (*S. aromaticum*) EO, and of citral, containing geranial (trans-citral, citral A) and neral (cis-citral, citral B), found in the citrus plants leaves and fruits. Eugenol acts by disrupting cellular membrane permeability, while citral affects both the cytoplasmic/outer membrane as well as the stress response mediated by the sigma factor RpoSin *E. coli* [43].

The *Thymbra capitata* EO inhibited the preformed biofilms of different *Candida* spp. at 2xMIC, excepting *C. glabrata*, probably due to the increased content in phenols (carvacrol) [44]. Starting from these observations on antifungal activity of EOs, Dalleau et al. (2008) have tried to deepen this study by testing ten terpenes, the main components of EOs (carvacrol, citral, eucalyptol, eugenol, farnesol, geraniol, linalool, menthol, γ-terpinene, and thymol), on different *Candida* strains (*C. albicans*, *C. parapsilosis*, *C. glabrata*). The best activity was recorded using carvacrol against *C. albicans*, *C. glabrata* and *C.parapsilosis* biofilms, the effect being biofilm age and concentration-independent. They also obtained good results for geraniol and thymol [45]. In another study [46], the antibiofilm effect of *Citrus limon* and *Zingiber officinale* EOs were investigated, and it has been shown that they can be used against biofilms of *Klebsiella ornithinolytica*, *K. oxytoca* and *K. terrigena*.

In an interesting article from 2019, Kerekeset al. described the antibiotic effect of *Cinnamomum zeylanicum*, *Origanum majorana*, and *Thymus vulgaris* EOs on dual-species biofilms.

Studying the effect on the biofilm produced by *L. monocytogenes* SZMC 21307 and *E. coli* SZMC 0582, they found that treatment with cinnamon EO at concentrations of 1 mg/mL eradicated the dual biofilm. In the case of marjoram EO, the biofilm elimination started from 0.5 mg/mL concentration, and in the case of thyme EO, the inhibitory effect was detected starting with 1 mg/mL concentrations. These values were surprisingly much lower than those recorded in the eradication of monoculture biofilms. All studied

EOs decreased biofilm formation but at concentrations higher than those required for monospecific biofilms eradication.

These polymicrobial biofilms can be found in the food industry, and the recorded results suggest the possible use of EOs as food preservatives, but however, their use is limited by the strong odor and taste, requiring further study to mitigate these effects [47].

The promising results on the antibiofilm effect of EOs have outlined a new challenge for researchers to study whether the association of EOs with antibiotics is beneficial. In this regard, Rosato et al. published in 2020 a series of research that is intended to be just the beginning of a comprehensive study on the synergistic effect of EO antibiotics. They studied the activity of *Cinnamomum zeylanicum*, *Mentha piperita*, *Origanum vulgare* and *Thymus vulgaris* EOs associated with norfloxacin, oxacillin, and gentamicin on bacterial biofilm produced by *S. aureus*, *S. epidermidis* IG4, and *E. faecalis*. The synergistic effects were tested through the checkerboard microdilution method. The study showed that all EOs have a synergistic effect, the best being in combination with norfloxacin, leading, for example, in the case of *Cinnamomum zeylanicum* EO to a decrease from 128 μg/mL to 3.99 g/mL of gentamicin MIC50.The advantages of combined therapy are obvious: the decrease of antibiotic doses and implicitly reducing the resistance to antimicrobial drugs [48].

*3.5. EOs Used in Dentistry*

Many studies were devoted to finding new irrigants or interappointment to remove the microbial biofilms formed in the mouth, which prevent endodontic treatments [49]. Therefore, there is a need for chemical substances as medications that have both antibacterial and antibiofilm activities. *E. faecalis* is commonly recovered from teeth with persistent endodontic infections, creating biofilms attached to the canal walls or located in isthmuses and ramifications from where are difficult to eliminate by current substances, such as sodium hypochlorite and chlorhexidine [41,50]. Microbial biofilms and smear layer must be eradicated during endodontic treatment. Because the substances used as chemical irrigants are not bio-friendly with the dental and peri-radicular tissues, different natural substances have been studied as disinfectantsof root canals [51]. Chloroformic solutions of eucalyptus and orange EOs associated with cetrimide at concentrations varying from 0.05% to 0.3% reduced the biofilm by 70–85%. The two EOs enhanced the efficiency of cetrimide, which effectively eradicated the biofilms in lower doses, the synergic effect being probably due to lipophilic compounds (e.g., terpenoids or phenolics) [50].

The *Melaleuca alternifolia* EO used as a gel with antibacterial effect was very effective against oral *S. mutans* biofilm, decreasing the gingival bleeding index. Mouthwashes with this EO have also decreased not only *S. mutans* but also the total oral bacteria counts. The EO was used in 5% concentration, which was well accepted, without side effects [52,53].

Testing several oral disinfectants, including those containing a mixture of *Aloe vera* and TTO, Smith et al. (2013) demonstrated that none of the mouthwashes effectively eradicated biofilms formed from oral and bloodstream isolates MRSA.The antibiofilm effect can be improved by increasing the concentration and exposure time [54]. Carvacrol and oregano oil were also the subjects of the research study undertaken by Nostro in 2007. Both of them are known for their effect on *Staphylococcus* strains; they showed in vitro effects on staphylococcal biofilms, the biofilm inhibitory concentrations values have registered at 2–4xCMI values. Dental plaque biofilm plays an essential role in oral pathology, the etiology of dental caries, but also in contamination of dental materials surfaces, such as those used in the implant–prosthetic rehabilitation (implants, impression materials, alloys for prosthetic use, etc.) [55]. Due to the biofilm matrix destabilizing effect, thymol is used in the mouthwash with anti-plaque effects [56].

Cortelli et al. (2014) noted how important could be the use of EOs (menthol, thymol, and eucalyptol) for oral health by preventing the biofilm formation in patients with prostheses [57]. In several cases, the EOs can be more efficient than cetylpyridinium chloride [58]. Haas et al. suggested that EOs can be used daily, in the long-term, for reducing

the supragingival plaque and, thus, gingivitis [59]. Quintas et al. have shown that EOs prevented de novo plaque-like biofilm development for 7 h after mouthwash, representing a possible alternative to chlorhexidine for the pre-surgical rinse or after periodontitis treatments [60,61]. In a study from 2013, Erriu et al. demonstrated that the mouthwash containing EOs compounds, such as eucalyptol, methyl salicylate, menthol and thymol, combined with ethanol, exhibits an improved antibiofilm activity at high dilution. The nonalcoholic mixture of EOs tested on *Aggregatibacter actinomycetemcomitans* strains had better anti-planktonic behavior [62]. It was also demonstrated the benefit of association of these EOs with xylitol in mouthrinse against *S. mutans*-derived biofilms, independent of the type of treatment or age of biofilm. This is a very promising treatment for the treatment and prevention of caries [63].The *Mentha piperita* and *Rosmarinus officinalis* EOs proved to be effective against *S. mutans*, one of the main agents of dental caries. Of the *Mentha piperita* EO, having a menthol concentration below 3.6% was more effectivethan rosemary oil(containing piperitone as the main component) and chlorhexidine (at 4000 and 8000 ppm). The use of toothpaste blended with EOsindicated that lower concentrations of the EOs were more effective than chlorhexidine [64]. The association of chlorhexidine with EO is indicated for better antibiofilm activity in oral treatment [65].

Eugenol and citral could represent better alternatives to chlorhexidine because, at subinhibitory concentrations, they are affecting biofilm formation and virulence of methicillin-susceptible *S. aureus*, MRSA and *L. monocytogenes* strains, having a low-risk for selecting resistance [66].

Bersan et al. studied the EOs (1 mg/mL) from twenty medicinal and aromatic species on biofilms produced in vitro by different microbial strains and compared the results with nystatin and chlorhexidine digluconate. The *Aloysia gratissima* and *Coriandrum* spp. EOs have strongly inhibited *C. albicans*, *Fusobacterium nucleatum*, *P. gingivalis*, *S. mitis* and *S. sanguis*. The *C. articulates* EO inhibited *F. nucleatum* and *P. gingivalis* biofilms. *A. gratissima* (1 mg/mL or 9% concentration) inhibited the *S. mitis* biofilm more intensively than chlorhexidine [67]. EOs, stannous fluoride and hexetidine associated with methylparaben and propylparaben decreased the in vitro peri-implant biofilm mass and activity by 39% to 56% and decreased gingivitis by 59% after continuous application. These EOs have also been shown to reduce the release of bacterial endotoxins and pathogenicity [68].

In a recent study, Marinković et al. studied the antibiofilm efficacy of *Cymbopogon martini* and *Thymus zygis* EOs on the multispecific biofilms of *S. mitis*, *S. sanguinis* and *E. faecalis* in the root canals of extracted teeth. They found that the addition of an oil-based irrigant to 1.5% sodium hypochlorite proved to be more efficient against biofilm development [69]. On the other side, exposure to biocides (e.g., triclosan) can increase the *S. mutans* hydrophobicity, increasing its susceptibility to EOs. Therefore, a combination of triclosan-containing toothpaste with EOS-based mouthrinse could reduce the acidic bacteria [70]. A good antibiofilm and anti-caries effect, comparable to that of chlorhexidine (0.12%), was observed for a mouth rinse containing *Matricaria chamomilla* L. EO (PerioGard®–Palmolive). The antibiofilm effect has been evaluated as a decrease in Colony-forming units (CFUs) for total *S. mutans*, *S. sobrinus* and *Lactobacillus* sp., and the anti-caries effect has been studied as the effect on enamel demineralization compared to phosphate-buffered saline (PBS) solution. The authors found a mineral loss reduction by 39.4% in the case of mouthwash containing EO from *Matricaria chamomilla* L., very close to that of chlorhexidine (47.4%). The authors considered that the experimental product having Chamomile EO significantly reduced enamel demineralization [71]. The *Mentha spicata* essential oil was tested for in vitro and in vivo antimicrobial and biofilm activities on *S. mutans* [72].

These results suggest that rosemary EO is efficient against cariogenic oral streptococci. Due to its major compound, eugenol, clove oil is a potent fungicidal, bactericidal and natural anesthetic compound. The *Eucalyptus* EO, reach in eucalyptol, showed antibiofilm activity against *C. albicans* biofilms [73]. The EOs from *A. gratissima*, *Baccharis dracunculifolia*, *C. sativum*, and *Lippia sidoides* demonstrated a potent inhibitory activity on *S. mutans* biofilm, probably due to the presence of thymol, carvacrol, and trans-nerolidol [74]. Mouthwashes

containing *Citrus hystrix* leaf EO alone or in combination with chlorhexidine inhibited the periodontopathogenic bacteria and *S. sanguinis* and *S. mutans* biofilms [75]. The EO of *B. dracunculifolia* has been studied for use in dental care because it is known that this EO inhibits the growth of *S. mutans*. This EO reduced the rate of biofilm after one week of use, at the same level as triclosan, being a good candidate to be implemented in new material for dental care [76]. *Curcuma longa* EO (0.5 to 4 mg/mL) inhibited the growth, acid production and *S. mutans* adherence to saliva-coated hydroxyapatite beads and biofilm development [77]. The EO extracted from seeds of the *Butia capitata* tree was tested on biofilms produced by aciduric bacteria, lactobacilli, and *S. mutans*, comparing with three commercial self-etching adhesives, and it was demonstrated that they were equally effective against tested microorganisms [78].

The *Coriandrum sativum* EOs exhibited an inhibitory activity against *C. albicans* oral isolates from patients with a periodontal disease, similar to nystatin, suggesting its promising potential for the prophylaxis and treatment of oral candidiasis [79]. The *Citrus limonum* and *C. aurantium* EOs exhibited an antibiofilm effect comparable to 0.2% chlorhexidine but lower than 1% sodium hypochlorite on multispecific biofilms formed by *C. albicans*, *E. faecalis* and *E. coli* [80]. *C. sativum* EO isprobably active through its major compounds (decanal and trans-2-decenal) that could bind membrane ergosterol, acting similarly to nystatin and amphotericin B. *C. articulatus* contains α-pinene that could interfere with cellular envelopes integrity, respiratory chain and ion transport *A. gratissima* and *L. sidoides* EOs were bactericidal and inhibited the production extracellular polysaccharides in *S. mutans* [81]. The *C. sativum* EO also inhibited the proteolytic activity of *C. albicans* and affected the normal morphology of yeast cells (at 156.0 to 312.50 mg/mL concentration), probably by affecting the membrane permeability, due to the presence of mono- and sesquiterpene hydrocarbons [82].

### 3.6. EOs in Chronic Wound Infection Treatment

Wounds chronicity is often associated with biofilm development. Farnesol and xylitol exhibited a significant inhibitory effect against *E. faecalis* biofilms, being, therefore, proposed as adjuvants for the treatment of chronic wound infections or caries [83]. Notably, Anghel et al. demonstrated the benefit of using a modified wound dressing nanofunctionalized with magnetite nanoparticles with sustained release of *S. hortensis* EO (rich in phenolic compounds, such as thymol, carvacrol, and para-cymene) against *C. albicans* biofilm [84].

### 3.7. EOs in the BAIs Treatment

BAIs are a major cause of morbidity and mortality in hospitalized patients. The treatment of biofilm-mediated infections requires the development of new antibiofilm strategies, which represent new scientific challenges.

When Park et al. (2007) tested the antibacterial effect of TTO on silicone tympanostomy tubes, they found that all tested concentration of EO (100%, 50%, 10% in tween) produced a reduced bacterial adherence of all MRSA strains, which may be explained by the alteration of adherence factors present on the bacterial cell surface. The bacterial cultures were obtained from otorrhea in patients with chronic suppurative otitis media, and the antibiofilm effect of TTO was evaluated in comparison to vancomycin. The MRSA exhibited a similar susceptibility to 50% TTO and vancomycin. The authors proposed TTO as an alternative for pediatric MRSA otorrhea treatment with tympanostomy tubes [85]. Brady et al. studied the *S. aureus* biofilm, formed on a cochlear implant resistant to all conventional antimicrobials, but 5% TTO completely eradicated it in one hour [86]. Recently, Malic et al. studied the antimicrobial activities of TTO, comparatively to terpinene, eucalyptol (1,8-cineole) and eugenol against *p. mirabilis* involved in catheter-associated urinary tract infections, for further use to obtain modified catheter biomaterials, but they found a reduced antibiofilm activity [87]. Previously, a study from 2010 showed that the pomelo EO inhibited the *S. epidermidis* and *P. aeruginosa* biofilms development on soft contact lenses in a time and temperature-dependent manner [88]. In their study, Selim

et al. presented the *Cupressus sempervirens* EO inhibitory effect on *K. pneumoniae* cells adherence capacity to intravenous infusion tubes made of polyvinyl chloride (PVC) at 500 μg concentration. It was observed that biomaterial surface pretreatment with EO rendered it repellent the microbial cells, thereby reducing surface adhesion [12]. *C. citratus* (at 0.5× and 0.25× MIC) and *Syzygium aromaticum* EOs inhibited the *C. albicans* clinical and reference strains biofilms formed under static conditions in polystyrene tubes [89,90], proposing them as an alternative to amphotericin B and fluconazole [91]. Cinnamon bark EO (containing as major components cinnamaldehyde and eugenol) has been shown to exhibit a potent antibiofilm on *P. aeruginosa*, reducing it by up to 96%. When mixed with 2% poly(D,L-lactide-co-glycolide) (PLGA) (a biodegradable polymer), it prevented *p* biofilm formation [92]. In their paper, Chmit et al. reported that they could not determine a notable antibiofilm effect of the EO from *Laurus nobilis* using a *S. epidermidis* strain. Despite these results, the *L. nobilis* EO remains in attention due to its large spectrum of activity against pathogenic bacteria [93,94]. The medical industry is in a constant search for new materials, but also of biocidal products, the potential applications of EOs and polymer systems attracting the attention of researchers. Nostro et al. incorporated eugenol, citronellol and linalool in poly(ethylene-co-vinyl acetate) copolymer (EVA) and tested against *E. coli*, *P. aeruginosa*, *L. monocytogenes*, *S. epidermidis*, and *S. aureus*. The EO diffused through the polymeric matrix, and the combinations EVA + citronellol and EVA + eugenol at 7% concentration induced a 40–90% biofilm decrease [95].

*3.8. EOs Used in Food Industry*

In the food industry, bacteria adhere to vats, tanks and tubes, impairing food safety and quality. Therefore, strategies are needed to inhibit biofilm formation or the elimination of mature biofilms. Current strategies used in the food industry, such as disinfection, surface preconditioning, ultrasonication, etc., although effective, cannot control microbial biofilms. The quorum-sensing systems that assure a coordinate gene expression depending on cellular density also regulating biofilm formation represent a promising lead for the development of novel antibiofilm strategies [96]. The food-contact surfaces rise many problems to the meat industry because of the risk of contamination with pathogenic (e.g., *Salmonella enterica*, *L. monocytogenes*, *E. coli*) ormeat spoilage bacteria (e.g., *Pseudomonas* spp., *Brochothrix thermosphacta* and *Lactobacillus* spp.) bacteria, predominantly growing in biofilms. Disinfection of the food contact surfaces is a difficult and challenging problem that can be solved by finding new disinfectants, such as EOs [6,97,98]. To find new food preservatives, detergents and sanitizers, which can be used in the food industry, Chorianopoulos et al. tested the *S. thymbra* EO (1% *v/v*) against monospecific or polyspecific biofilms formed by Gram-positive (*S. simulans*, *L. fermentum*, *L. monocytogenes*) or Gram-negative (*P. putida*, *S. enterica*) bacteria. The strong inhibitory effect of EO on microorganisms was associated with carvacrol and thymol compounds, acting as membrane permeabilizing agents [99]. *Mentha piperita*, *C. citratus* and *Cinnamomum zeylanicum* EOs inhibited *S. enterica*-serotype *Enteritidis* biofilm development on stainless steel surfaces, for the first two EOs, after 10–40 min [100]. Sub-MIC concentrations of cinnamon EO and cinnamaldehyde reduced the biofilm counts at 156–234 μg/mL. Cinnamaldehyde is probably acting by inhibiting the macromolecules synthesis and damaging the cell membrane [101]. The *C. citratus* EO was also tested comparing with the *T. vulgaris* EO against *A. hydrophila* biofilm development on stainless steel coupons in UHT skimmed milk [11]. The *Thymbra capitata* EO was evaluated against both planktonic and biofilm cells of *S. enterica* serovar *Typhimurium* and proved very efficient in comparison with benzalkonium chloride [102]. One of the proposed alternatives to overcome the bacterial contamination of the food, both in suspension and in the biofilm, can be the use of a mixture of essential components of volatile oils (thymol, eugenol, berberine and cinnamaldehyde), to create a synergistic effect with streptomycin, useful in controlling foodborne pathogens [103]. The antimicrobial activity of 19 EOs was evaluated to determine their effectiveness in eliminating the pathogenic agent *S. aureus* from the food processing plants. Thus, planktonic cells of *S. aureus* strains have shown

increased sensitivity to volatile thyme, lemon and vetiver oils. The 48 h old biofilms of the same strains, formed on stainless steel installations in the food industry, could not be eliminated by the volatile oils tested, with increased efficiency of oils of thyme and patchouli. The efficiency of thyme oil was increased by the use of benzalkonium chloride. To prevent the emergence of resistant strains, it is necessary to combine different types of essential oils, as well as their use in combination with various biocides [32].

### 3.9. EOs as Air Disinfectants

Laird et al. examined how Citri-V, a mixture of citrus EOs (orange: bergamot, 1:1 $v/v$), removes *Enterococcus* spp. and *S. aureus* biofilms from then stainless steel and plastic surfaces after aerial release in a concentration of 15 mg/L. The citrus vapors may be used for additional decontamination if they are also used with routine cleaning, as they are less toxic than ozone or hydrogen peroxide used for air decontamination treatments [104].

### 3.10. Nanoparticles with EOs Used in Controlling and Preventing Infections

Nanoparticles used in antibiofilm therapy have been intensively studied lately due to their unique properties, helping to fight resistant infections because they can easily penetrate the biofilm matrix and can also functionalize biomedical surfaces by coating, impregnating or embedding, thus preventing biofilm formation [105].

The use of functionalized nanoparticles with EOs can be a method for controlling and preventing infections associated with microbial biofilms, limiting at the same time the consumption of synthetic antimicrobial drugs and thus, reducing microbial resistance.

Nanoparticles can be used ascontrolled and local delivery systems for EOs and also for enhancing their activity. The development of combinations between nanoparticles and EOs is a new research direction approached also by Chifiriuc et al., who used *Rosmarinus officinalis* EO to obtain a nanobiosystem used for coating catheter surface that successfully inhibited the adherence of *C. tropicalis* and *C. albicans* clinical strains. After 48–72 h, the biofilm was almost absent on the surface of the coated materials [106].

As we also presented in this paper, many studies have shown the beneficial effect of carvacrol in inhibiting microbial biofilms, attracting the attention of researchers in the field of nanotechnology, which have tried to encapsulate this EO in PLGA nanocapsules to obtain a new drug delivery system, that altered the architecture of the *S. epidermidis* biofilm when added in the initial phases of biofilm formation [107].

The aim of a study performed by Bilcu and coworkers figured out the characteristics resulted from combining the antimicrobial activity of three EOs obtained from *Pogostemon cablin*, *Vanilla planifolia* and *Cananga odorata* subsp. with that of iron oxide@C14 nanoparticles for obtaining coatings for the surfaces of medical devices. These hybrid coatings inhibited the *S. aureus* and *K. pneumoniae* adherence and biofilm formation in both initial and maturation (in the case of vanilla EO) phases [108].

Polylactic acid (PLA) nanoparticles with EOs can be used to design new ecological strategies based on natural alternatives efficiently in the treatment of severe infections with biofilms formed by pathogenic and/or resistant bacteria. PLA was combined with lemon EO to obtain functional nanocapsules that exhibited better antimicrobial activity than PLA alone [109]. Using a solvent evaporation method and coated with matrix-assisted pulsed laser evaporation (MAPLE), a hybrid nanocoating composed ofmagnetite nanoparticles functionalized with *Melissa officinalis* EO, PLA and chitosan, was obtained and characterized. In vitro experiments revealed significant inhibitory activity of prokaryotic cell adhesion properties [110]. New research has been effectuated on the development of nanoemulsions containing geranium oil that demonstrated inhibitory effect against *C. albicans*, *C. tropicalis*, *C. glabrata* and *C. krusei* biofilms, quantified by measuring the total protein and bioluminescence. The results showed a better activity on *C. albicans* and *C. tropicalis* biofilms. Both geranium oil and nanoemulsions containing this oil significantly inhibited biofilm formation in all species tested on polyethylene surfaces, with nanoemulsions having a better activity, proving that they can become an effective alternative for reducing the

microbial adhesion on the surface of the medical devices and preventing consecutive infections [111]. For increasing its thermal stability, carvacrol is incorporated into polymers of the type of halloysite nanotubes, which then allow subsequent mixing with the low-density polyethylene melt. The nanocomposites exhibited antimicrobial activity against *E. coli*, *L. innocua* and *Alternaria alternate* biofilms. They have proven effective and can be excellent candidates for a wide range of applications, such as controlling microbial contamination of food [112].

**Table 1.** The EOs studied in terms of microbial antibiofilm action.

| Latin Name of Plant Source of EO | Main Components of EO | Microbial Strain That Produces Biofilms on Which the EO Has Been Tested | Reference |
|---|---|---|---|
| *Boswellia papyrifera* *Boswellia rivae* | n-octyl acetate, octanol, limonene, a-pinene, verticilla-4 (20), 7,11-triene, acetate, incensole | *Staphylococcus epidermidis* *S. aureus* *C. albicans* | [31] |
| *Butia capitata* | capric, caprylic, lauric, linoleic, myristic, oleic, palmitic, stearic acids | *Aciduric bacteria* *Lactobacilli* *Streptococcus mutans* | [78] |
| *Cananga odorata* subsp. *Genuine* (ylang-ylang oil) | *p*-cresyl methyl ether, linalool, geranyl acetate, geraniol, eucalyptol | *S. aureus* *Klebsiella pneumoniae* | [109] |
| *Cinnamomum aromaticum* (Cassia oil) | cinnamaldehyde, eugenol, linalool | *P. aeruginosa* *P. putida* *S. aureus* | [7] |
| *Cinnamomum zeylanicum* | e-cinnamaldehyde | *Salmonella* Saintpaul | [102] |
| *Citrus hystrix* | citronellal | *S. sanguinis* *S. mutans* | [75] |
| *Coriandrum sativum* | decanal, trans-2-decenal, 2-decen-1-ol, cyclodecane | *C. albicans* *C. tropicalis* *C. krusei* *C. dubliniensis* *C. rugosa* | [82] |
| *Cupressus sempervirens* | $\alpha$-pinene, $\alpha$-terpinolene, $\delta$-3-carene, limonene | *K.pneumoniae* | [12] |
| *Curcuma longa* | curlone, *trans-β*-elemenone, germacrone, $\beta$-sesquiphellandrene, $\alpha$-turmerone, $\alpha r$-turmerone, $\alpha$-zingiberene | *S. mutans* | [77] |
| *Cymbopogon citratus* (lemongrass oil) | geranial, neral, myrcene | *C. albicans* *C. tropicalis* *C. glabrata* *C. krusei* *P. gingivalis* *P. intermedia* *Aeromonas hydrophila* | [89–91] [73] [11] |
| *Eucalyptus camaldulensis* | eucalyptol | *Porphyromonas gingivalis* *Actinobacillus actinomycetemcomitans* *Fusobacterium nucleatum* *S. mutans* *S. sobrinus* *C. albicans* | [73] |

**Table 1.** *Cont.*

| Latin Name of Plant Source of EO | Main Components of EO | Microbial Strain That Produces Biofilms on Which the EO Has Been Tested | Reference |
|---|---|---|---|
| *Eugenia caryophyllata (Syzygium aromaticum)* (clove oil) | biflorin, caryophyllene oxide, eugenol, eugenyl acetate, ellagic acid, gallic acid, kaempferol, myricetin, oleanolic acid, rhamnocitrin | *C. albicans* *C. tropicalis* *C. glabrata* *C. krusei* *P. gingivalis* *P. intermedia* | [73,89–91] |
| *Laurus nobilis* | acetate, eucalyptol, linalool, methyleugenol, *α*-terpinyl | *S. epidermidis* | [93,94] |
| *Lavandula angustifolia* (lavender essential oil, LEO) | camphor, caryophyllene, eucalyptol, lavendulyl acetate, limonene, linalool, linalyl acetate, *cis*-ocimene, 3-octanone, a-pinene, transocimene, terpinen-4-ol | *S. aureus* *E. coli* | [5] |
| *Matricaria chamomilla* (Chamomile EO) | (E)-β-farnesene α-bisabolol oxide A | *S. mutans* *S. sobrinus* *Lactobacillus* sp. | [71] |
| *Melaleuca alternifolia* (tea tree oil, TTO) | *α*-pinene, *p*-cymene, eucalyptol,terpinen-4-ol, *γ*-terpinene,*α*-terpinene, terpinolene | *S. mutans* *Proteus mirabilis* | [52] [87] |
| *Melissa officinalis* (Melissa essential oil, MEO or lemon balm) | citrals (geranial + neral, citronellal, limonene, geraniol, β-caryophyllene, β-caryophyllene oxide, and germacrene D | *S. aureus* *E. coli* | [5] |
| *Mentha piperita* | menthofuran, menthol, menthyl acetate, eucalyptol, menthone, α-pinene, sabinene, β-pinene | *C. albicans* *C. dubliniensis* | [34] |
| *Mentha spicata* | carvone, trans-carveol, myrcenecarvyl-acetate-Z | *S. mutans* | [73,98] |
| *Ocimum gratissimum* | eugenol, 1,8-cineole | *S. aureus* *E. coli* | [13] |
| *Origanum vulgare* (oregano oil) | carvacrol, thymol, γ-terpinene, *p*-cymene | *S. aureus* *S. epidermidis* | [55] |
| *Pogostemon cablin* (patchouli essential oil) | α-guaiene, β-caryophyllene, δ-cadinene, pogostol, (-)-patchoulol, seychellene, α- and β-patchoulene | *S. aureus* *K. pneumoniae* | [109] |
| *Rosmarinus officinalis* (rosemary oil) | eucalyptol, alpha-pinene, camphor, verbenone, borneol | *S. sobrinus* | [73] |
| *Satureja thymbra* | carvacrol, thymol, *p*-cymene | *S. simulans* *Lactobacillus fermentum* *P. putida* *Salmonella enterica* *Listeria monocytogenes* | [6] |
| *Thymbra capitata* | carvacrol, γ-terpinene, *p*-cymene | *Candida albicans* *C. glabrata* *C. tropicalis* *C. parapsilosis* *C. guilliermondii* | [44] |
| *Thymus vulgaris* (thyme oil) | eucalyptol, camphor | *A. hydrophila* | [11] |
| *Vanilla planifolia* (vanilla oil) | ethylvanillin, 4-hydroxybenzaldehyde, methyl anisate, 4-hydroxybenzyl methyl ether, piperonal, vanillic acid, vanillin | *S. aureus* *K. pneumoniae* | [109] |

**Table 2.** The EOs studied in terms of microbial antibiofilm action formed on a medical device or live surface.

| Latin Name of Plant Source of EO | The Support (Medical Device) on Which the Biofilm Was Studied | Reference |
|---|---|---|
| *Baccharis dracunculifolia* | Dental biofilm | [76] |
| *Mentha spicata* | | [73] |
| *Melaleuca alternifolia* | | [52] |
| *Cananga odorata* subsp. *genuine* | Catheter | [109] |
| *Pogostemoncablin* | | |
| *Vanilla planifolia* | | |
| *Cupressus sempervirens* | Intravenous infusion tube | [12] |
| *Cymbopogon citratus* | Stainless steel coupons | [11] |
| *Thymus vulgaris* | | [11] |
| *Saturejathymbra* | | [6] |
| *Eugenia caryophyllata* (*Syzygiumaromaticum*) | Soft contact lenses | [33] |
| *Thymus vulgaris* | | |
| *Lavandula angustifolia* | Urological catheter, infusion tube, surgical mesh | [5] |
| *Melissa officinalis* | | |
| *Melaleuca alternifolia* | | |
| *Melaleuca alternifolia* | Catheter-associated urinary tract infections | [87] |
| | Cochlear implant | [86] |
| | Silicone tympanostomy tubes | [85] |

## 4. Conclusions

Essential oils represent safe and efficient alternatives for the development of novel antibiofilm agents that could find a potential role in the medical and food industries for infection control, especially BAIs associated with artificial medical devices, susceptible to the formation of microbial films resistant to conventional antibiotic treatment. The great advantage of EOs is that their usage is not likely to select for microbial resistance because they have a complex composition and, therefore, multiple targets in the microbial cells.

An increasing idea to combat microbial biofilms is to combine the conventional antimicrobials with EOs, based on studies already conducted, which have shown promising results.

Our review shows that most studies were performed in vitro, thus further in vivo studies are necessary, as well as the elucidation of many additional therapeutic aspects, such as EOs formulation, frequency and duration of therapy, safety issues; these aspects need to be optimized to ensure the best possible clinical outcomes.

It cannot be neglected the potential advantage of using EOs prophylactically, and in this context, a promising lead is to obtain bioactive nanobiocoatings containing EOs for inhibiting bacterial and fungal adhesion and further biofilm development on the different surface from the medical, industrial and natural environment.

**Author Contributions:** Conceptualization, D.C.N. and C.L.; methodology, I.Z.; software, C.C. and I.N.; investigation, T.C. and P.I.; data curation, D.C.N., C.L., M.C.C. and I.Z.; writing—original draft preparation, D.C.N., C.L., C.C., M.C.C., T.C., P.I., I.N. and I.Z.; writing—review and editing, D.C.N., M.C.C. and P.I.; visualization, T.C.; supervision, D.C.N., C.L. and I.Z. All authors have read and agreed to the published version of the manuscript.

**Funding:** This research was funded by UEFISCDI, PN-III-P1-1.1-TE-2019-1506, project number 147TE/2020.

**Institutional Review Board Statement:** Not applicable.

**Informed Consent Statement:** Not applicable.

**Data Availability Statement:** Not applicable.

**Conflicts of Interest:** The authors declare no conflict of interest.

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
