# Peer review of "Contribution of Essential Oils to the Fight against Microbial Biofilms—A Review"

_processes, doi:10.3390/pr9030537_

Round 1

Reviewer 1 Report

This review provides a qualified overview on EO application to counter the biofilm development

Author Response

Many thanks to the reviewer for appreciating our article. We made some changes suggested by the other reviewers

Reviewer 2 Report

The manuscript basicly is well written with some points to be addressed.

  • Chamomile oil is missing from the list although it has significant antimicrobial property
  • There is no description of the applied administration methods of the EOs (topical, inhalation, etc.)
  • The MIC values should be listed in a table
  • The measuring systems are not described. This is extremely important because the hydrophobicity is a major problem in aqueous test systems
  • The mode of action of EOs is poorly described. Many EOs induce oxidative stress and metabolic alterations in the microbes
  • It would be important to mention some examples on the synergistic action of EOs in combination with standard antimicrobials
  • The anti-biofilm action is even more difficult to measure, the authors should describe some accepted test systems
  • The English of the paper should be rechecked

Author Response

We want to thank the reviewer for the valuable remarks that we took into consideration in correcting the article. We hope that the additions we have made respond to the observations and increase the scientific level of our paper.

  • Point 1: The action of chamomile EO is missing, it also has strong antimicrobial effect.

Response 1: We thank the reviewer for the critical remark. We added the effect of this essential oil. It is also presented in Table 1.

  • Point 2: The various EO formulations are not given. This is an important issue because EOs are very hydrophobic and are difficult both to apply and to measure their antimicrobial efficiency in aqueous environment. The different test systems should be listed with special emphasis on biofilms.

Response 2: Indeed, this is an important aspect that we did not approach in the article. We have presented some recent research that facilitates the testing of the antimicrobial effect of EOs.

  • Point 3: There is no mention on the various molecular explanations on the action of EOs only the cell wall effects are described. Many authors prove that EOs induce oxidative damage and also inhibit metabolic and synthetic pathways in the microbes.

Response 3: Thank you for this observation, we completed with data about other types of mechanisms of action (oxidative stress, enzymatic action)

  • Point 4: The route of administration is also important (topical, inhalation, etc.).

Response 4: We agree, but this aspect is difficult to complete, because the research is mostly done in vitro, except for mouthrinses used to control the dental microbial biofilms. We filled in with the data found in the literature

  • Point 5: It would be nice to have a table that lists the MIC values for the EOs shown in the paper.

Response 5: You have right, but we could not make such a table, because the research is not unitary, neither in terms of the studied EOs nor the type of microbial strains that cause biofilms.

  • Point 6: It would also increase the strength of the work if the authors could demonstrate synergistic actions of EOs administered together with standard antibiotic and/or antifungal agents.

Response 6: We thank the reviewer for this interesting suggestion, we complete with the information that we found important. The research on the association of EOs with antibiotics to combat microbial films is at the beginning, and data from the literature are few compared to those on synergistic antimicrobial action.

Reviewer 3 Report

The review  very thoroughly describes the usage of essential oils (EOs) as antimicrobial agents with special focus on anti-biofilm actions. In general the manuscript is well written, however I list some shortages that should be corrected.

  • The action of chamomile EO is missing, it also has strong antimicrobial effect.
  • The various EO formulations are not given. This is an important issue because EOs are very hydrophobic and are difficult both to apply and to measure their antimicrobial efficiency in aqueous environment.
  • The different test systems should be listed with special emphasis on biofilms.
  • There is no mention on the various molecular explanations on the action of EOs only the cell wall effects are described. Many authors prove that EOs induce oxidative damage and also inhibit metabolic and synthetic pathways in the microbes.
  • The route of administration is also important (topical, inhalation, etc.).
  • It would be nice to have a table that lists the MIC values for the EOs shown in the paper.
  • It would also increase the strength of the work if the authors could demonstrate synergistic actions of EOs administered together with standard antibiotic and/or antifungal agents.

Author Response

We want to thank the reviewer for the valuable remarks that we took into consideration in correcting the article. We hope that the additions we have made respond to the observations and increase the scientific level of our paper.

We have made the changes suggested by all reviewers and we hope that this time our article fulfils the necessary criteria for publication.

Point 1: The research summary provided in this manuscript is too broad that lost focus when it flow towards the end. The objective of the review must be clarified initially and should explain the same in detail.

Response 1: We thank the reviewer for the critical remark. The Abstract was changed.

Point 2: Typo- and repetitious text found. 

Response 2: Thank you, we have tried to correct these mistakes

Point 3: Table 2 need modifications. Eg. dental biofilm was given at three different places which should be categorized under one

Response 3: We agree with the reviewer; we have deleted the mentioned rows from Table 2. We organized the table as you have suggested.

Point 4: Entire structure of this article needs improvement. There are many broken paragraphs without continuity and vague explanations.

Response 4: Thank you, we have changed.

Point 5: The entire MS need an expert proof-read before considering for further processes. I would recommend rejection and resubmission.

Response 5: We agree with the reviewer; we have done changes, suggested by all reviewers

Point 6: Get help with the language.

Response 6: We apologize for our mistakes, we have corrected the text

Reviewer 4 Report

  1. The research summary provided in this manuscript is too broad that lost focus when it flow towards the end. The objective of the review must be clarified initially and should explain the same in detail. 
  2. Typo- and repetitious text found.  
  3. Table 2 need modifications. Eg. dental biofilm was given at three different places which should be categorized under one.
  4. Entire structure of this article needs improvement. There are many broken paragraphs without continuity and vague explanations.
  5. The entire MS need an expert proof-read before considering for further processes. 
    I would recommend rejection and resubmission.

Author Response

We want to thank the reviewer for the valuable remarks that we took into consideration in correcting the article. We hope that the additions we have made respond to the observations and increase the scientific level of our paper.

Point 1: The abstract mentions "prostheses, common dental implants, heart valves, etc". Microbial biofilm may present a problem for such devices, but to the extent that the devices are internal it is difficult to see how an essential oil would be delivered. The section on essential oil use in dentistry does include information on peri-implant biofilm. It is suggested that heart valves be removed.

Response 1: We thank the reviewer for the critical remark. We removed heart valves from the text

Point 2: Throughout: author and col. --> author et al.

Response 2: Thank you, we have changed.

Point 3: Is xylitol an essential oil component? If not, expand the figure legend.

Response 3: We removed the xylitol from the figure.

Point 4: lines 151 - 152: The first sentence in this paragraph should be at the end of the introduction.

Response 4: Thank you, we have changed.

Point 5: When you define an abbreviation at the first occurrence of the full term use the abbreviation subsequently.  At line 153 you define tea tree oil as TTO.  At line 157 it is again defined.

Response 5: We agree with the reviewer; we have done the modifications.

Point 6: Get help with the language.

Response 6: We apologize for our mistakes, we have corrected the text

Reviewer 5 Report

1. The abstract mentions "prostheses, common dental implants, heart valves, etc". Microbial biofilm may present a problem for such devices, but to the extent that the devices are internal it is difficult to see how an essential oil would be delivered. The section on essential oil use in dentistry does include information on peri-implant biofilm. It is suggested that heart valves be removed.

2. Throughout: author and col. --> author et al.

3. Figure 1. Is xylitol an essential oil component? If not, expand the figure legend.

4. lines 151 - 152: The first sentence in this paragraph should be at the end of the introduction.

5.  When you define an abbreviation at the first occurrence of the full term use the abbreviation subsequently.  At line 153 you define tea tree oil as TTO.  At line 157 it is again defined.

6. Get help with the language.

Author Response

There is no reviewer 5

Round 2

Reviewer 2 Report

The authors responded to every issues properly, have made the necessary changes in the manuscript and rephrased the English language correctly.

Reviewer 4 Report

The revised manuscript is acceptable for publication.